# Diagnosis and Management of Catastrophic Antiphospholipid Syndrome and the Potential Impact of the 2023 ACR/EULAR Antiphospholipid Syndrome Classification Criteria

**DOI:** 10.3390/antib13010021

**Published:** 2024-03-12

**Authors:** Lucas Jacobs, Nader Wauters, Yahya Lablad, Johann Morelle, Maxime Taghavi

**Affiliations:** 1Internal Medicine Department, Brugmann University Hospital, Université Libre de Bruxelles, 1020 Brussels, Belgium; 2Nephrology and Dialysis Department, Brugmann University Hospital, Université Libre de Bruxelles, 1020 Brussels, Belgium; 3Internal Medicine Department, Tivoli University Hospital, Université Libre de Bruxelles, 7100 La Louvière, Belgium; 4Division of Nephrology, Namur University Hospitals (CHU UCL Namur), 5000 Namur, Belgium

**Keywords:** anticardiolipin, antiphospholipid antibodies, antiphospholipid syndrome, catastrophic antiphospholipid syndrome, lupus anticoagulant, sepsis, systemic lupus erythematosus, thrombotic microangiopathy

## Abstract

Catastrophic antiphospholipid syndrome (CAPS) is a rare and life-threatening condition characterized by the persistence of antiphospholipid antibodies and occurrence of multiple vascular occlusive events. CAPS currently remains a diagnostic challenge and requires urgent treatment. The diagnosis of CAPS is made difficult by classification criteria used as diagnostic criteria in clinical practice, knowledge derived from retrospective data and case reports, confounding clinical and biological features, and its rapid onset and mortality. The absence of prospective studies of CAPS limits the strength of evidence for guideline treatment protocols. This comprehensive review summarizes the current understanding of the disease, and discusses how the 2023 ACR/EULAR Antiphospholipid Syndrome Classification Criteria impact the definition and therapeutic management of CAPS, which is considered the most severe form of APS. The correct integration of 2023 ACR/EULAR APS classification criteria is poised to facilitate CAPS diagnosis, particularly in critical situations, offering a promising avenue for improved outcomes.

## 1. Introduction

Catastrophic antiphospholipid syndrome (CAPS) is a rare and severe form of antiphospholipid syndrome (APS) associated with high mortality. It is more commonly encountered in intensive care units (ICU), and may sometimes be confused with other serious conditions presenting similar clinical and biological phenotypes. In acute scenarios, promptly diagnosing CAPS poses a considerable challenge due to the complexity of the interpretation of antiphospholipid antibody (aPL) assays, the non-specific nature of its clinical and biological presentation, and the low sensitivity of available criteria, which can contribute to misdiagnosis. Furthermore, the extreme rarity of CAPS results in clinicians often lacking awareness of the disease. 

Developing evidence-based clinical guidelines for rare diseases like CAPS poses significant challenges due to the scarcity of primary evidence, absence of randomized trials, hesitation in formulating weak recommendations that may hinder treatment funding, and a restricted pool of patients for engagement [1]. Therapeutic options are, therefore, heterogeneous and supported by a low degree of evidence because of the absence of clinical studies. Treatment is thus often initiated on a probabilistic basis, considering a convergence of supportive evidence and a careful risk–benefit assessment.

In this article, we focused on CAPS, its definition, pathophysiology, diagnostic challenges, and treatments in acute clinical practice. We connected CAPS knowledge with the new classification criteria for APS released in September 2023.

This article constitutes a literature review, conducted by searching the PubMed database for the term ‘catastrophic antiphospholipid syndrome’ from its first usage in 1992 by R. Asherson [2] until November 2023. 

## 2. Overview of Antiphospholipid Syndrome

APS is an autoimmune disease first described in the 1980s, characterized by vascular thrombosis and/or obstetric morbidity in the presence of persistent positivity (defined as two positive tests at a minimum of 12 weeks apart) for at least one of the antiphospholipid antibodies (aPLs) [3]. 

The overall annual incidence of APS is approximately five per 100,000 inhabitants, varying significantly with age, gender, and country of origin [4]. APS can be isolated, referred to as “primary”, or secondary to an autoimmune disease. Systemic lupus erythematosus (SLE) is the most commonly associated autoimmune disease [5]. In the Europhospholipid cohort, which included 1000 APS patients, 53% had primary APS, while 36% had APS associated with SLE, with lupus-like syndromes in 5% and with other diseases in 6% [6]. 

The three main laboratory tests commonly performed in clinical practice are the lupus anticoagulant (LA) test (using a functional coagulation test), anti-beta-2 glycoprotein 1 antibodies (aβ2GP1) test, and anticardiolipin antibodies (aCL) test (using immunoassay techniques). The persistent positivity of one or more aPL in association with arterial, venous, and microvascular thrombotic manifestations, and/or obstetric complications defines APS. Non-thrombotic manifestations, formerly termed ‘non-criteria manifestations’, are also associated with APS [7].

Until recently, the 2006 revised Sapporo classification criteria were used to diagnose APS [8] (Table 1). Since then, advancements in our understanding of APS include better characterization of aPL-associated, non-thrombotic clinical manifestations, identification of the role of traditional thrombosis risk factors in aPL-positive individuals, and risk stratification by aPL profile. Furthermore, the revised Sapporo criteria did not incorporate certain evidence-based definitions (e.g., microvascular disease or pregnancy morbidity), resulting in the inclusion of a heterogeneous group of “aPL-positive” patients with different risk profiles for research [3].

The new 2023 APS classification criteria [3] issued by the American College of Rheumatology (ACR) and the European Alliance of Associations for Rheumatology (EULAR) include certain clinical and biological entities previously considered as “non-criteria manifestations” of APS, including skin, kidney, heart, and hematologic complications of the syndrome. The 2023 ACR/EULAR classification also proposes a stratification of macrovascular thrombotic risk through the assessment of traditional venous thromboembolism and cardiovascular disease risk factors with weighted assessment. For instance, in a patient with a high thromboembolic risk profile, the occurrence of thrombosis carries a lower diagnostic weight compared to an episode of thrombosis in a patient with no additional thrombotic risk. The 2023 ACR/EULAR criteria also bring forward well-defined microvascular domain items thought to be mechanistically distinct from moderate-to-large vessel disease, restructured definitions of pregnancy morbidity, and the addition of cardiac valve disease and thrombocytopenia. Table 1 summarizes the differences between the new 2023 ACR/EULAR classification criteria and the 2006 revised Sapporo classification criteria for APS. 

It is important to note that both the modified Sapporo criteria and the 2023 ACR/EULAR criteria are classification criteria aimed at including a homogeneous population of patients with a certain APS diagnosis in clinical studies. While these criteria have high specificity (few false positives), their sensitivity is slightly lower. The absence of these criteria does not exclude the diagnosis, but they are often used in clinical practice, with clinicians being mindful of their limitations and referring complex cases to expert centers.

## 3. Definition and Epidemiology of CAPS

The CAPS Registry, established in 2000 by the European Forum on Antiphospholipid Antibodies, is a key tool that has significantly contributed to our current understanding of CAPS. The registry includes over 500 cases classified as definite (meeting all classification criteria) and probable (treated as such despite not meeting all criteria) [10] (Table 2).

CAPS was defined by Ronald Asherson in 1992 [2] as a condition characterized by multiple vascular occlusive events (defined as at least three distinct events), typically affecting small vessels, over a short period (defined as ≤7 days), confirmed by histopathology, and the persistent presence of aPL over 12 weeks, usually at high titers [11,12]. The 2023 ACR/EULAR APS classification is likely to contribute to the revision of CAPS classification criteria, providing a better understanding of APS and its various clinical and biological phenotypes.

**Table 2 antibodies-13-00021-t002:** Definitions of definite and probable CAPS.

	Criteria
Definite CAPS (All four criteria present)	1. Evidence of involvement of three or more organs, systems, and/or tissues
	2. Development of manifestations simultaneously or in less than a week
	3. Histopathological confirmation of small vessel occlusion in ≥1 organ/tissue
	4. Laboratory confirmation of the presence of antiphospholipid antibodies ≥12 weeks
Probable CAPS (One of the following)	All four criteria, but only two organs, systems, and/or sites of tissue involvement
	All four criteria, except for the laboratory confirmation at least 6 weeks apart due to the early death of a patient never tested for aPL before catastrophic APS
	Criteria 1, 2, and 4 above
	1, 3, and 4, and the development of a third event in more than a week but less than a month, despite anticoagulation

CAPS: catastrophic antiphospholipid syndrome; antiphospholipid antibodies (aPL): lupus anticoagulant, anticardiolipin antibodies, and/or anti-beta-2 glycoprotein 1. Adapted from [12].

CAPS accounts for only 1% of reported APS cases [6], making systematic study challenging. In the overall CAPS Registry cohort, 60% had primary APS, while 30% were observed in patients with secondary APS [10,13]. We will refer to primary CAPS in cases where CAPS occurs in patients with primary APS, and to “SLE-associated CAPS” in cases where CAPS occurs in those with SLE-associated APS. In the literature, “secondary CAPS” is often mentioned, referring to CAPS in cases of secondary APS. Given that CAPS in patients with APS secondary to an autoimmune disease other than SLE are exceptional, the term “secondary CAPS” usually refers to “CAPS in SLE-associated APS patients”.

In early published series [14], the mortality rate was 50%. However, this rate has decreased year by year [15]. Despite aggressive treatment and a better understanding of this syndrome, CAPS mortality still stands at 37%, including both primary CAPS and SLE-associated CAPS, with SLE-associated CAPS mortality approaching 48% despite earlier diagnosis [10]. The precise overall incidence of CAPS, due to its rarity and diagnostic challenges, is likely still unclear to date.

## 4. Pathophysiology and Triggering Factors of CAPS

The pathophysiology of CAPS remains partially understood to date. As implied by the “triple-hit” hypothesis, additional pro-thrombotic factors should be present to transform a state of hypercoagulability into CAPS [16]. The pathophysiology is complex and multifactorial. It involves aPL stimulating endothelial cells as well as immune cells through interactions with anionic determinants, facilitated by the β2GP1 cofactor. This results in a pro-coagulant state [16,17], an inhibition of fibrinolytic systems [18,19], neutrophil activation, neutrophil extracellular traps (NETosis) [20,21], and activation of signaling pathways such as nuclear factor-κB (NF-κB) which is associated with a pro-inflammatory and pro-coagulant profile of endothelial cells [21,22]. This leads to the expression of adhesion molecules and the production of cytokines, promoting inflammation and thrombosis. Furthermore, aPLs contribute to the decreased activity of endothelial nitric oxide synthase, impair vasodilation, and promote platelet adhesion [23].

In addition, platelet activation, adhesion, and aggregation occur [17], and macrophages, neutrophils, and other cellular actors are also implicated, likely along with some as-yet unidentified genetic factors [23,24]. Data suggest the activation of both the classic and alternative pathways of complement in CAPS [25,26,27,28,29]. It may be explained by the higher rate of variants in genes encoding for proteins regulating the alternative complement pathway in patients with CAPS [25]. Furthermore, CAPS manifestations could be related to systemic inflammatory response syndrome, resulting from the excessive release of cytokines into the blood by affected and necrotic tissues [20]. It is now recognized that SIRS can result from both sepsis and non-infectious causes, such as immune-mediated organ injury [19]. Acute respiratory distress syndrome, encephalopathy, and myocardial dysfunction, common clinical manifestations in CAPS patients, have all been described in cases of systemic inflammatory response syndrome [18,30,31]. The administration of corticosteroids leading to reduced cytokine production [32] may explain its effectiveness in CAPS [22]. 

New in vitro studies are focusing on the role of IgG aβ2GP1 subclasses and Fc glycosylation in the pathogenesis of CAPS. Indeed, aβ2GP1 IgG of CAPS patients displayed a prominent increase of bisection, agalactosylated glycopeptides, fucosylation, and sialylation, which alluded to a pathogenic role of aβ2GP1 IgG glycosylation in the pathogenesis of APS, and a plausible role as markers of ensuing CAPS. Core fucosylation of aβ2GP1 IgG1/2 subclasses was strongly associated with CAPS and triple positivity of antiphospholipid antibodies. These changes can alter autoantibody functions, such as affecting C1q binding, C5a-mediated inflammation, and antibody-dependent cell cytotoxicity in CAPS. Indeed, altered IgG Fc N-glycosylation of aβ2GP1 displays a pro-inflammatory pattern in APS [33]. 

Due to its extreme rarity and the lack of in vitro/in vivo models, the pathophysiology of CAPS remains poorly understood to date. The reason for the lack of studies on the pathophysiological mechanisms of CAPS is mainly attributed to challenges in collecting serum samples during acute episodes. These challenges stem from the condition’s low prevalence, complexities in distinguishing it from other microangiopathic conditions, and its high mortality rate [34]. Most pathophysiological hypotheses stem from retrospective data from the CAPS Registry, as well as extrapolation from the physiopathology of APS. 

A specific characteristic of CAPS is that 60% of patients appear to have one or more triggering factors, especially infections, present in 30–50% of cases [15,35] (Appendix A). Other identified precipitating factors include pregnancy (22%), surgical interventions or trauma (17%), neoplasms (16%), contraceptives (10%), or the discontinuation of anticoagulation [10,35,36]. Several triggering factors may be present in the same patient (e.g., withdrawal of anticoagulation followed by surgical intervention or biopsy in patients with neoplasia and aPL antibodies). Underlying mutations in complement regulatory genes could serve as a “second hit”, leading to uncontrolled complement activation and a more severe thrombotic phenotype [25]. This ‘double-‘ or ‘triple-hit’ hypothesis may apply to any patient with multi-organ failure [20]. Due to its diagnostic challenges in acute situations, the precise study of the precipitating factors of CAPS is likely still imperfect to date. 

## 5. Clinical and Biological Manifestations of CAPS

CAPS differs from classical APS in that it primarily affects small vessels, with thrombotic events occurring simultaneously or over a short period and at multiple different sites. Classical APS, on the other hand, predominantly affects large vessels, although not exclusively, with thrombotic events typically being sporadic and limited to a single site.

CAPS is primarily characterized by diffuse thrombotic microangiopathy with a predilection for the kidneys, lungs, brain, heart, skin, and gastrointestinal tract [10]. Specifically, the clinical presentation of CAPS is characterized by kidney involvement in 73% of cases, with varying degrees of kidney dysfunction, and pulmonary involvement in 60% of cases, presenting as acute respiratory distress syndrome or pulmonary embolisms (26%) [37]. Up to 56% of patients exhibit cerebral manifestations, such as stroke or encephalopathy [10,38]. The heart is affected in half of cases, primarily as myocardial infarction or valvulopathy. Libman–Sacks endocarditis is reported in 13% of CAPS with cardiac involvement, especially in SLE-associated CAPS [39]. These patients also show a high frequency of clinical manifestations of cytokine storm and microangiopathic phenomena, clinically expressed by a higher frequency of livedo and Raynaud’s syndrome [13]. Purpura is a less frequent manifestation of CAPS (see Figure 1) [10,40]. Severe HELLP syndrome (Hemolysis, Elevated Liver enzymes, and Low Platelets) appears to be a major feature of CAPS during the obstetric period, which is already susceptible to thrombotic events [41,42]. Other rarer symptoms are illustrated in Figure 1. 

There is significant variability in clinical presentations and precipitating factors related to the age of patients and the primary or secondary status of CAPS [13]. In children, CAPS is more frequently associated with peripheral venous thrombosis (37% vs. 23% in elderly adults) and heart failure (37% vs. 19%), while in the elderly, CAPS is more frequently associated with arterial thrombosis (33% vs. 16%) and renal involvement (87% vs. 71%) [10,43].

Recurrent CAPS is rare and corresponds to the recurrence, more than 30 days after apparent clinical and hematological remission, of clinical and laboratory findings compatible with CAPS [44]. This has been reported in less than 5% of CAPS cases [10,45].

Biologically, according to the CAPS Registry, LA was found in 83% of proven CAPS cases, aCL IgG in 81%, aCL IgM in 49%, anti-β2-GP1 IgG in 78%, and anti-β2GP1 IgM in 40% of cases [10] (Appendix A). Low plasma levels of plasma C3 and C4 fractions of complement were detected in 58% of patients with CAPS and have been inconsistently associated with clinical presentation, pro-thrombotic state, or higher mortality [25,46]. Biological evidence of thrombotic microangiopathy was found in a third of cases [10].

Current knowledge about the clinical manifestations of CAPS mainly comes from retrospective data collected in the CAPS Registry and from case series or case reports. Given the diagnostic challenges of CAPS cases, it is likely that we underestimate its incidence and, consequently, miss certain clinical manifestations.

## 6. CAPS in SLE-APS Patients

SLE-associated CAPS is frequent, represents one-third of the cases of CAPS, and predominantly affects women, with a female-to-male ratio of 8:1 [13]. Given the high prevalence of a history of SLE, clinical suspicion of CAPS is mandatory. This prevalence might increase in the future. Indeed, the new ACR/EULAR 2023 criteria could lead to an overdiagnosis of APS in the setting of SLE given that thrombocytopenia and aPL positivity are commonly seen in SLE patients and are integrated into the 2019 EULAR/ACR classification criteria for SLE (80). Primary APS and SLE overlap in clinical and immunological features. Misclassification of APS as SLE is a concern and remains a challenge. The 2019 ACR/EULAR SLE classification criteria have a high accuracy for distinguishing APS from SLE; however, misclassification has been reported [47]. 

According to the registry, SLE flare represents a precipitating factor of CAPS. Also, CAPS mortality is significantly higher in lupus patients (48% vs. 33% for primary CAPS). The presence of SLE in CAPS patients is a prognostic factor for mortality after adjusting for age, sex, organ involvement, and treatment [13]. SLE-associated CAPS occurs on average earlier and CAPS can be the mode of presentation of a lupus flare [10]. This may be attributable to the early onset of SLE and/or an increased incidence of thrombosis in SLE patients independent of antiphospholipid antibodies [48]. In the Europhospholipid cohort, patients with SLE-associated CAPS more frequently presented with arthritis and livedo [6]. In the CAPS Registry, patients with SLE-associated CAPS had more cardiac, neurological, and cutaneous involvement, and more thrombocytopenia. Interestingly, antinuclear antibody positivity in APS patients should be acknowledged as a risk factor for potential relapses irrespective of a clinical diagnosis of connective tissue disease. In this specific cluster, clinical and biological work-ups should focus on microvascular features and/or cytopenias, especially considering a propensity for CAPS [49]. 

## 7. Meeting Diagnostic Challenges

In 2002, the Task Force on Catastrophic Antiphospholipid Syndrome developed criteria identifying definite and probable CAPS with a sensitivity of 89% and specificity of 100% [50]. Later, at the 13th International Congress on Antiphospholipid Antibodies in 2010, Erkan et al. developed diagnostic algorithms to assist clinicians in confirming their diagnoses [51]. These algorithms attempt to reconcile CAPS criteria with different clinical scenarios. Indeed, the diagnosis of CAPS is challenging in clinical practice due to its non-specific biological markers in the acute phase and its classification criteria erroneously used as diagnostic criteria, which are not well-suited for emergency situations. In fact, “definite CAPS” would represent only 41% of described cases [52]. Table 2 presents the current definitions for definite and probable CAPS.

The rarity and rapid onset of CAPS impede enrollment in prospective studies, resulting in epidemiological data primarily sourced from retrospective data. Retrospective knowledge around clinical manifestations of CAPS is mainly gathered in the CAPS Registry, representing the largest cohort of published cases. However, the CAPS Registry describes a heterogeneous cohort of patients with CAPS by different groups leading to difficulty in organizing the information and probably leading to some information bias. Also, the latter study suffers publication bias since the cohort is established mostly based on case reports or series. 

The first clinical practice difficulty concerns the interpretation of positivity or negativity of antiphospholipid antibodies (aPL). Firstly, an aPL test can be falsely positive during infection or inflammatory syndrome (usually a positive LA or a low-titer aPL ELISA test), or anticoagulant treatment (usually a positive LA test) [51,53,54]. Conversely, falsely negative aPL titers can occur during acute thrombotic events [55,56]. A common scenario is the initial detection of a positive LA test and/or low-titer aPL ELISA test simultaneously with the diagnosis of multivisceral thrombosis, in the presence or absence of thrombosis risk factors (infections, being postoperative, disseminated intravascular coagulation, etc.). 

To date, CAPS diagnosis relies on double testing of aPL antibodies 12 weeks apart and histological evidence of microvascular thrombosis. However, in approximately half of cases, CAPS occurs in patients without a history of previous antiphospholipid syndrome. The *McMaster RARE-Bestpractices clinical practice guideline* for the diagnosis and management of CAPS suggests using classification criteria as a diagnostic tool for CAPS (with very-low-level evidence) and relying on aPL positivity [1]. These recommendations caution against the risk of false negatives, especially aPL, which can lead to treatment discontinuation. The *McMaster RARE-Bestpractices clinical practice guideline* suggests performing a biopsy, which, if positive, would be highly specific evidence for CAPS (very-low-level evidence). Nevertheless, in an acute context, performing a biopsy is often difficult, and one of the main reasons for being unable to define CAPS as definite rather than probable [51]. A strong suspicion of CAPS should be based on clinical features of multivisceral involvement, particularly when associated with clinical evidence suggestive of microangiopathic vasculopathy [34]. The reliance on clinical expertise for accurate diagnosis poses challenges. The overall incidence and prevalence of CAPS is, therefore, likely underestimated.

Therefore, upon reasonable suspicion of CAPS (e.g., presence of two classification criteria), it is advisable to initiate empirical treatment, weighing the benefits and risks, and to confirm the diagnosis through histological analyses, allowing confirmation or refutation of the diagnosis, sometimes after the event. 

## 8. Therapeutic Management of CAPS 

### 8.1. First-Line Treatments

The treatment guidelines of CAPS were established more than 15 years ago, and remain poorly standardized to date, given the lack of comparative studies and randomized trials due to the extremely low incidence of this disease, lack of randomized trials, reluctance to make ‘weak’ recommendations that could jeopardize the chance of getting treatment funding, and limited patients available for involvement [1]. This results in significant heterogeneity in management practices from one hospital center to another that could bring heterogeneity in patients’ outcomes [10]. 

The cornerstone of CAPS treatment is the association of therapeutic anticoagulation, corticosteroids, and treatment of the condition that triggered CAPS (if identified).

Anticoagulation with unfractionated heparin has the most significant impact on the patient’s life prognosis [10,57,58]. Heparin also appears to have anti-inflammatory and complement-inhibitory effects in CAPS [59]. Vitamin K antagonists, initiated after heparin, are long-term anticoagulants [60], given that treating thrombotic APS patients with direct oral anticoagulants compared with a vitamin K antagonist appears to increase the risk for arterial thrombosis [61]. However, there appears to be insufficient data in the current literature to study subgroups according to anticoagulant used.

Corticosteroids are usually administered in intravenous pulses at doses of 0.5–1 mg/kg, followed by tapering to achieve a short treatment duration (typically 4–6 weeks). This represents a very strong experts’ recommendation with low-level evidence as the use of a glucocorticoid is supported by observational evidence of benefit when combined with other therapies [62].

Therapeutic plasma exchange is an established procedure in CAPS, especially for patients with microangiopathic features or renal involvement, despite a lack of evidence regarding prescription details, including frequency, choice of replacement fluid, timing, and number of sessions. Typically, clinicians perform daily sessions for 2 to 3 weeks [1,57,63,64,65]. Monitoring the treatment response is advisable to assess when to stop the exchanges (e.g., normalization of platelet count or the absence of schistocytes on the blood film). It has been suggested that titrating antiphospholipid antibodies (aPL) could be a reliable marker of response to plasmapheresis [66] (low-level evidence).

Intravenous immunoglobulins (IVIG) have been described as beneficial in primary CAPS, particularly in cases of immune thrombocytopenia [10,13]. However, there have been supply difficulties for several years due to an imbalance between demand (increased indications for treatment) and production. IVIG could be useful in patients with hemodynamic instability, difficult vascular access for plasma exchanges, and cases of severe/immune thrombocytopenia with a significant risk of bleeding. No difference has been observed between regimens involving plasma exchanges or IVIG [62]. 

All CAPS considered, the highest survival rate has been achieved with a combination of anticoagulation, corticosteroids, and plasma exchanges (72%). The highest survival rates were reached in patients with SLE-associated CAPS who received anticoagulation, corticosteroids, and plasma exchanges (65%), and in patients with primary CAPS who received anticoagulation, corticosteroids, and IVIG (82%) [13]. Thus, as soon as the diagnosis is suspected, patients should receive anticoagulation and corticosteroids (first-line therapies) [67], along with plasma exchanges, and/or IVIG [1,12,68]. To date, no treatment modality has been prospectively studied in a randomized manner. Therefore, any recommendation represents low-level evidence and expert opinions. Furthermore, there are no studies evaluating the harmful effects or costs of these combinations of therapies in CAPS.

### 8.2. Add-on Therapies

In case of clinical deterioration, an add-on therapy may be considered (see Figure 2) [69]. Cyclophosphamide may constitute an effective additional therapy in the presence of SLE, especially if active lupus manifestations are present. However, cyclophosphamide could confer an unfavorable prognosis in primary CAPS [13]. 

The *McMaster RARE-Bestpractices clinical practice guideline* suggests using antiplatelet agents as an add-on therapy (a conditional recommendation with a very low certainty of evidence). In patients for whom anticoagulation is contraindicated for reasons other than bleeding, using antiplatelet agents as an alternative is strongly recommended (very low certainty of evidence) [1].

Rituximab could improve patients’ outcomes, although its use is only described in about twenty patients [70,71,72]. There is no consensus on the best dosing regimen [10,71,73]. These recommendations represent low-level evidence and expert opinions. Indeed, in a review of 20 out of 441 (4.6%) patients included in the “CAPS Registry” who were treated with rituximab, 16 (80%) patients recovered from the acute CAPS episode and 4 (20%) died at the time of the event. However, there is high heterogeneity with respect to the first/second-line treatment and the reason for initiation. Also, the isolated effect of rituximab is difficult to analyze because of the low number of patients and given the fact that all these patients received a combined triple therapy [34,71].

### 8.3. Refractory CAPS

Confirmation of complement involvement in the pathophysiology of APS [74] and CAPS would justify the use of eculizumab in patients refractory to conventional treatment, also considering its rapid action (low-level evidence) [69,75,76,77]. It has been suggested that eculizumab could also be useful in preventing CAPS, for example, before transplantation (very-low-level evidence) [78]. However, evidence regarding the use, efficacy, efficiency, and safety of eculizumab is extremely low and limited to case reports.

Sirolimus inhibits the mTOR signaling pathway, a therapeutic target in vascular lesions associated with APS nephropathy [79,80,81], and appears promising in CAPS, but there is currently insufficient data to recommend its use.

Other therapies have been mentioned, such as defibrotide, which modulates the activity of TNF-α, endothelin, thrombin, and IL-2. It also has a beneficial effect on endothelial dysfunction, demonstrating antithrombotic, anti-ischemic, and anti-inflammatory effects, without anticoagulant activity [82]. It also has a demonstrated an anti-NETosis effect in APS [67]. Unfortunately, data regarding the efficacy or safety of defibrotide in CAPS are limited to two patients with a 50% survival rate [82]. 

### 8.4. SLE-Associated CAPS Treatment

Because of the poorer prognosis of CAPS in the setting of SLE, a first-line add-on therapy could be a therapeutic option to improve prognosis. Despite low-level evidence, cyclophosphamide may constitute an effective additional therapy in the presence of SLE since it has been associated with a lower mortality rate in a multivariate analysis of the data included in the “CAPS Registry” [13,34]. Cyclophosphamide is recommended in CAPS associated with SLE. Rituximab may also be an option in SLE patients [83].

In refractory CAPS associated with active SLE, in addition to CYC or rituximab, hydroxychloroquine would decrease platelet activation, reduce attachment of the aPL-β2GP1 complex to the membrane, and decrease thrombotic and cardiovascular risk [82]

Other treatments, well documented in SLE (e.g., belimumab, an anti-BAFF human monoclonal antibody, anifrolumab, a monoclonal antibody against type I interferon receptor subunit 1, etc.) could also present potential therapeutic options in SLE-associated CAPS [84]. However, the efficacy and safety profiles of these drugs have never been evaluated in this population.

## 9. Impact of 2023 ACR/EULAR Criteria on CAPS Patients

The 2023 ACR/EULAR classification criteria mark a significant milestone since the previous revised Sapporo criteria in 2006. These criteria have been validated in two cohorts with 99% specificity, with a slightly lower sensitivity [3], and may soon impact CAPS diagnosis, epidemiology, presentation, and prognosis. 

The new APS classification criteria ACR/EULAR 2023, with their clinical domains suggestive of thrombotic phenomena (notably the microvascular domain items), could facilitate the diagnosis of CAPS in critical situations. Indeed, the inclusion of “microvascular domain items” (e.g., thrombocytopenia, cardiac valve involvement, and livedo racemosa) in the 2023 ACR/EULAR classification criteria, allows for the recruitment of patients excluded by the Sapporo criteria, reclassifying some as “APS”.

On the contrary, some individuals with a history of high thrombotic risk, despite testing positive for aPL antibodies and experiencing arterial/venous thromboses, will no longer be classified as APS. Additionally, patients testing positive only for aCL and/or aβ2GP1 IgM, regardless of the titer, will not meet the new criteria for APS despite the fact that those with double positivity for aCL and aβ2GP1 IgM are recognized to have a high-risk thrombotic profile [85]. The CAPS registry notes that IgM aCL accounts for 49% of aPL positivity, while IgM aβ2GP1 represents 40%, though the association of IgM aCL or IgM aβ2GP1 with double and triple positivity is not specified.

One study on 965 potential patients with APS among whom 436 were clinically diagnosed as APS by experienced rheumatologists, found that 18% of APS patients are reclassified as non-APS according to the new 2023 ACR/EULAR criteria. This is mostly because they exhibited only aCL and/or aβ2GP1 IgM positivity, resulting in low laboratory domain scores in 74% of patients. Also, 38% of these patients scored less than three points in clinical domains such as obstetrical complications. In this study only a few patients with thrombocytopenia, cardiac valve involvement, and livedo racemosa were reclassified as APS (*n* = 9) [85]. Also, the new ACR/EULAR 2023 criteria could lead to an overdiagnosis of APS in the setting of SLE given that thrombocytopenia and aPL positivity are commonly seen in SLE patients and are integrated into the 2019 EULAR/ACR classification criteria for SLE [86]. 

Finally, misuse of the new 2023 ACR/EULAR as diagnostic criteria rather than classification criteria could lead to a rise of CAPS cases. By no longer considering certain patients as having APS, doctors could potentially alter the chronic treatments of APS patients who would no longer meet the classification criteria. One can envision discontinuation of chronic anticoagulation in a patient positive for IgM aCL and/or aβ2GP1 and having experienced a thrombosis because they no longer meet the 2023 ACR/EULAR classification criteria. The discontinuation of anticoagulation in a patient with APS is a significant risk factor for the development of CAPS. 

Further studies are needed to assess whether these new APS classification criteria will have an impact on the epidemiology, clinical and laboratory presentations, and prognosis of CAPS. SLE subgroup patients as well as the patients with only IgM aCL and/or aβ2GP1 positivity will need particular attention for further studies.

## 10. Conclusions

CAPS is the most severe variant of APS. Its diagnosis is challenging in acute situations, where multiple pathologies may overlap with its clinical and biological manifestations. Its classification criteria, as well as its treatment, are mainly defined based on the pathophysiological knowledge of APS and retrospective studies, many of which are derived from the CAPS Registry. Prompt and aggressive treatment, ideally tailored to the primary or SLE-associated nature of CAPS, is key to the prognosis of the disease. The new APS classification criteria ACR/EULAR 2023, with their clinical domains suggestive of thrombotic phenomena, could facilitate the diagnosis of CAPS in critical situations, and will likely modify the epidemiology and presentation of CAPS. However, if misused as diagnostic criteria, the new 2023 ACR/EULAR could increase the risk of developing CAPS in some patients who are now considered non-APS. It is essential to exercise extreme caution and vigilance to avoid overlooking the disease in patients who have risk factors for CAPS. Continued investigation into the pathological mechanisms of CAPS and the involvement of thrombo-inflammation holds the potential to greatly enhance the prevention and treatment of this life-threatening condition. Such research could lead to the development of effective medications capable of disrupting pivotal pathological pathways.

## Figures and Tables

**Figure 1 antibodies-13-00021-f001:**
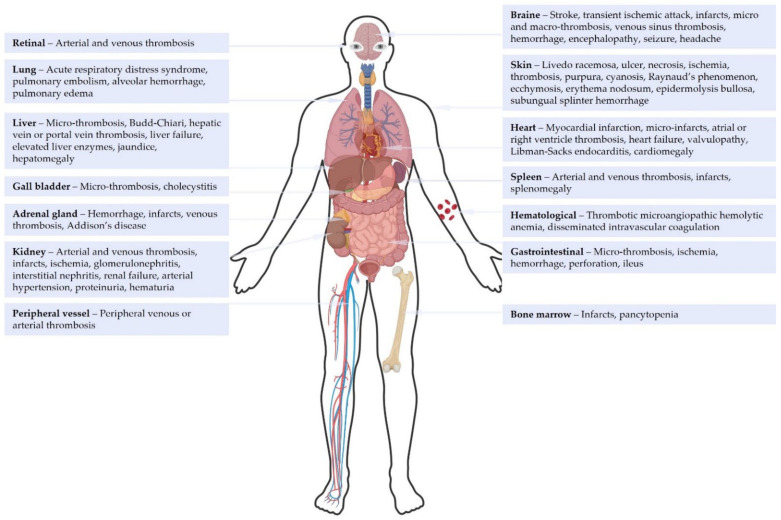
Clinical manifestations of catastrophic antiphospholipid syndrome according to the literature [10,13]. Figure 1 was created with BioRender.com.

**Figure 2 antibodies-13-00021-f002:**
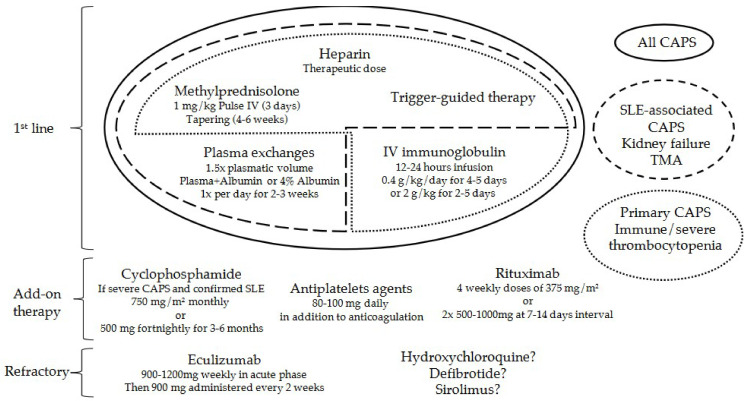
Overview of the available treatment options for catastrophic antiphospholipid syndrome. Treatment regimens are divided into first-line, add-on therapy, or treatment of refractory CAPS. CAPS: catastrophic antiphospholipid syndrome, IV: intravenous, SLE: systemic lupus erythematosus, TMA: thrombotic microangiopathy.

**Table 1 antibodies-13-00021-t001:** Comparison between new 2023 ACR/EULAR APS classification criteria and 2006 revised Sapporo classification criteria.

	2006 Revised Sapporo	2023 ACR/EULAR
**Classification**	≥1 clinical criteria AND ≥1 laboratory criteria	≥3 points from clinical domains AND ≥3 points from laboratory domains
**Clinical criteria**	Clinical criteria: 1. Vascular thrombosis: ≥1 clinical episode of arterial, venous, or microvascular thrombosis in any tissue or organ 2. Pregnancy morbidity	Clinical domains: 1. Macrovascular—venous thromboembolism * 2. Macrovascular—arterial thromboembolism * 3. Microvascular ** 4. Obstetric 5. Cardiac valve 6. Hematology
Included in the APS criteria		
Heart valve disease	No	Yes
Livedo racemosa	No	Yes
Thrombocytopenia	No	Yes
Nephropathy	No	Yes
Neurological manifestations	No	No
Pulmonary hemorrhage	No	Yes
Adrenal hemorrhage	No	Yes
**Laboratory criteria**		
Persistent positivity (at 12 weeks)	Yes	Yes
Timeline of aPL positivity and clinical criteria	Within 5 years of clinical criterion	Within 3 years of clinical criterion
Threshold of aCL and/or aβ2GP1	aCL > 40 GPL/MPL units, or >99th percentile aβ2GP1 > 99th percentile	aCL or aβ2GP1: Moderate: 40–79 units High: ≥80 units
Antibodies for laboratory criteria		
Positive LAC	Yes	Yes
IgG aCL or aβ2GP1	Yes	Yes
IgM and/or aβ2GP1	Yes	Yes (not sufficient if isolated)

* in the setting of high- or low-thrombotic risk profiles (diagnostic weight varying). ** proven histologically or clinically (diagnostic weight varying). aPL: antiphospholipid antibodies, aCL: anticardiolipin antibodies, aβ2GP1: anti-beta-2 glycoprotein 1 antibodies, GPL: IgG phospholipid unit, MPL: IgM phospholipid unit, LAC: lupus anticoagulant. Adapted from [9].

## Data Availability

No new data were created or analyzed in this study. Data sharing is not applicable to this article.

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
