# Peer review of "Diagnosis and Management of Catastrophic Antiphospholipid Syndrome and the Potential Impact of the 2023 ACR/EULAR Antiphospholipid Syndrome Classification Criteria"

_2073-4468, 2024, doi:10.3390/antib13010021_

Round 1
Reviewer 1 Report
Comments and Suggestions for Authors
Peer Review Report on "Diagnosis and Management of the Catastrophic Antiphospholipid Syndrome and Potential Impact of the 2023 ACR/EULAR Antiphospholipid Classification Criteria"
1. Summary of the Manuscript: The manuscript presents a comprehensive review of the catastrophic antiphospholipid syndrome (CAPS), focusing on its diagnosis, management, and the implications of the 2023 ACR/EULAR Antiphospholipid Syndrome Classification Criteria. The authors provide an overview of CAPS, discuss its pathophysiology, clinical manifestations, and therapeutic management, and evaluate the potential impact of the new classification criteria.
2. Major Concerns:
a. Abstract and Introduction (Sections Abstract and 1):
- The abstract, while informative, lacks a critical perspective on the limitations of current understanding and the criteria themselves, which is crucial for setting the stage for a comprehensive review​​.
- The introduction provides a general overview of CAPS but misses an opportunity to critically address the gaps in current research, which is essential for a review aiming to influence clinical practice and future research​​.
b. Content and Depth of Analysis (Sections 2, 3, 4, and 5):
- The discussion on APS and its association with CAPS is basic and lacks a critical evaluation of the limitations of current diagnostic criteria​​.
- The manuscript details the CAPS Registry's contributions but does not critically analyze potential biases or limitations inherent in registry data​​.
- Pathophysiology and triggering factors are described, yet the manuscript lacks an in-depth exploration of the uncertainties and gaps in current understanding​​.
- Clinical manifestations of CAPS are effectively described but the manuscript fails to critically analyze the limitations of current knowledge and the potential underestimation of CAPS incidence​​.
c. Specific Subgroups and Diagnostic Challenges (Sections 6 and 7):
- The significance of SLE in CAPS patients is highlighted, but the manuscript does not delve into the complexities of managing SLE-associated CAPS or the need for targeted research in this subgroup​​.
- While addressing diagnostic challenges, the manuscript falls short in critically evaluating the limitations of current diagnostic approaches​​.
d. Therapeutic Management (Section 8):
- The section on treatment options is comprehensive but lacks a critical analysis of the heterogeneity in management practices and the implications on patient outcomes​​.
- Add-on therapies are mentioned, but there is no critical evaluation of their efficacy or the evidence supporting their use​​.
e. Impact of New Criteria (Section 9):
- The potential impact of the new classification criteria is discussed, but the manuscript does not adequately examine the risks of misclassification and its consequences​​.
3. Conclusion: The conclusion provides a summary of the key points, but it does not effectively highlight the urgent need for more research, especially prospective studies, to improve the understanding and management of CAPS​​.
4. Overall Evaluation: The manuscript, while informative, requires substantial revisions to enhance its critical analysis and depth. The authors are encouraged to provide a more comprehensive critique of current knowledge gaps, diagnostic challenges, and the implications of the new classification criteria. There is a need for a stronger emphasis on the urgent requirement for more research and standardized approaches in CAPS diagnosis and management.
5. Recommendations:
- Revise the abstract and introduction to highlight the limitations and gaps in current understanding and research.
- Provide a more critical analysis of the APS overview, CAPS Registry data, pathophysiology, and clinical manifestations.
- Deepen the discussion on specific subgroups, particularly SLE-associated CAPS.
- Enhance the critical evaluation of therapeutic management and add-on therapies.
- Critically examine the potential impact and risks associated with the new classification criteria.
- Strengthen the conclusion with a clear call for more research and standardized approaches in CAPS management.
Reviewer's Decision: Major Revisions Required
Reviewer 2 Report
Comments and Suggestions for Authors
The present review focuses on the rare but life-threatening condition known as catastrophic antiphospholipid syndrome (CAPS). It is characterized by persistent antiphospholipid antibodies and the occurrence of multiple vascular occlusive events. The author examines the previous Sapporo classification of antiphospholipid syndrome (APS) patients and the new 2023 ACR/EULAR APS classification criteria. The author offers a summary of the current understanding of the disease and highlights how the 2023 ACR/EULAR Antiphospholipid Syndrome Classification Criteria impact the definition and therapeutic management of CAPS. The author also cautions that the misuse of the criteria could be detrimental to the treatment of certain APS patients. Overall, the manuscript is well-written, providing a comprehensive and informative description of the state of the art in APS and CAPS studies, with a particular focus on the therapeutic treatment based on the new 2023 ACR/EULAR Antiphospholipid Syndrome Classification. However, I would suggest some changes that could improve the manuscript
Major points:
Comment 1:
I would like to request a more detailed explanation of the IgG subclasses involved in APS and CAPS, along with their degree of glycosylation, as mentioned in the paper at https://doi.org/10.1093/rheumatology/keab416.
We are aware that different subclasses can trigger an immune response and activate the complement in various ways. Since different IgG subclasses can bind FcgRs with varying affinities, it is essential to discuss the potential differences in FcgR expression in patients. Additionally, the authors may want to consider including a more in-depth analysis of the various IgG subclasses as a potential criterion for the 2023 ACR/EULAR Antiphospholipid Syndrome Classification.
Minor points:
Comment 1:
Page 2, lines 52 – 55:
It would be nice to have a graphical representation of the APS as “primary or secondary to an autoimmune disease”.
Comment 2:
Table 1:
The bullet points without any consistent alignment are distracting! Please consider eliminating or aligning better.
Comment 3:
Paragraph “DEFINITION AND EPIDEMIOLOGY OF CAPS”:
We would appreciate a small table, also in the supplementary materials, so that readers can check it easily.
Comment 4:
Figure 1
The resolution of the figure is not high enough to read the different boxes very well. Please consider increasing it for the publication
Comments on the Quality of English LanguageThe manuscript's English level is high, but it could benefit from some shortening in several sentences.
Round 2
Reviewer 2 Report
Comments and Suggestions for Authors
Thank you to the Authors for the changes and brilliant work done to meet the requests of the reviewer.
Author Response
On behalf of all co-authors, I would like to further thank the reviewers for the quality of their comments.
Lucas Jacobs